# Spatiotemporal mosaic self-patterning of pluripotent stem cells using CRISPR interference

**Ashley RG Libby[1,2], David A Joy[1,3], Po-Lin So[1], Mohammad A Mandegar[1], Jonathon M Muncie[3,4], Federico N Mendoza-Camacho[1], Valerie M Weaver[4], Bruce R Conklin[1,5], Todd C McDevitt[1,6]\***

[1]Gladstone Institute of Cardiovascular Disease, San Francisco, United States; [2]Developmental and Stem Cell Biology Graduate Program, University of California San Francisco, San Francisco, United States; [3]Graduate Program in Bioengineering, University of California Berkeley, University of California San Francisco, San Francisco, United States; [4]Department of Surgery, Center for Bioengineering and Tissue Regeneration, University of California, San Francisco, United States; [5]Department of Medicine, Division of Genomic Medicine, University of California, San Francisco, United States; [6]Department of Bioengineering and Therapeutic Sciences, University of California, San Francisco, United States

**\*For correspondence:**
todd.mcdevitt@gladstone.ucsf.edu

**Abstract** Morphogenesis involves interactions of asymmetric cell populations to form complex multicellular patterns and structures comprised of distinct cell types. However, current methods to model morphogenic events lack control over cell-type co-emergence and offer little capability to selectively perturb specific cell subpopulations. Our *in vitro* system interrogates cell-cell interactions and multicellular organization within human induced pluripotent stem cell (hiPSC) colonies. We examined effects of induced mosaic knockdown of molecular regulators of cortical tension (ROCK1) and cell-cell adhesion (CDH1) with CRISPR interference. Mosaic knockdown of ROCK1 or CDH1 resulted in differential patterning within hiPSC colonies due to cellular self-organization, while retaining an epithelial pluripotent phenotype. Knockdown induction stimulates a transient wave of differential gene expression within the mixed populations that stabilized in coordination with observed self-organization. Mosaic patterning enables genetic interrogation of emergent multicellular properties, which can facilitate better understanding of the molecular pathways that regulate symmetry-breaking during morphogenesis.
DOI: https://doi.org/10.7554/eLife.36045.001

## Introduction

Early morphogenic tissue development requires the robust coordination of biochemical and biophysical signaling cues to dictate cell-cell communication, multicellular organization, and cell fate determination. (*Burdsal et al., 1993*; *Leckband et al., 2011*; *Montero and Heisenberg, 2004*). A hallmark of morphogenesis is the asymmetric co-emergence of distinct cell populations that self-organize to form developmental patterns, multicellular structures, and ultimately, functional tissues and organs (*Bronner, 2016*; *Lancaster and Knoblich, 2014*; *Sasai, 2013*). For example, during gastrulation, the blastocyst transitions from a relatively homogeneous population of pluripotent cells to a spatially organized, multicellular composition of distinct progenitor cells. Therefore, to study morphogenesis, it is essential to promote the coincident development of analogous heterogeneous populations *in vitro*. Human pluripotent stem cells (hPSCs) provide an unlimited source of cells that can

**eLife digest** Embryos begin as a collection of similar cells, which progress in stages to form a huge variety of cell types in particular arrangements. These patterns of cells give rise to the different tissues and organs that make up the body.

Although we often use 'model' organisms such as mice and frogs to study how embryos develop, our species has evolved unique ways to control organ development. Investigating these processes is difficult: we cannot experiment on human embryos, and our development is hard to recreate in test tubes. As a result, we do not fully understand how developing human cells specialize and organize.

Libby et al. have now created a new system to study how different genes control cell organization. The system uses human pluripotent stem cells – cells that have the ability to specialize into any type of cell. Some of the stem cells are modified using a technique called inducible CRISPR interference, which makes it possible to reduce the activity of certain genes in these cells.

Libby et al. used this technique to investigate how changes to the activity of two genes – called ROCK1 and CDH1 – affect how a mixed group of stem cells organized themselves. Cells that lacked ROCK1 formed bands near the edges of the group. Cells that lacked CDH1 segregated themselves from other cells, forming 'islands' inside the main group. The cells retained their ability to specialize into any type of cell after forming these patterns. However, specific groups of cells were more likely to become certain cell types.

The method developed by Libby et al. can be used to study a range of complex tissue development and cell organization processes. Future work could create human tissue model systems for research into human disease or drug development.

DOI: https://doi.org/10.7554/eLife.36045.002

mimic developmental differentiation processes and maintain the ability to self-organize into tissue-like structures, such as optic cups, gut organoids, or stratified cortical tissues (*Eiraku et al., 2008*; *Eiraku et al., 2011*; *Spence et al., 2011*). However, due to the intrinsic variability of organoids (*Bredenoord et al., 2017*) and the lack of alternative human models that faithfully promote asymmetric emergence, many of the mechanisms that control and coordinate morphogenesis remain undefined. Therefore, new approaches to reliably control the emergence and organization of multiple cell types would greatly advance tissue modeling and organ developmental studies.

Controlling cellular heterogeneity *in vitro* is often achieved by independent differentiation of hPSCs followed by re-combination of distinct cell types, which fails to mimic parallel cell-type emergence (*Matthys et al., 2016*). Attempts to engineer *in vitro* systems that yield controlled emergence of spatial organization often rely on extrinsic physical restriction of cells to direct subsequent multicellular pattern formation (*Hsiao et al., 2009*; *Warmflash et al., 2014*). Physical constraints allow the observational study of cell-cell interactions within defined regions, but artificially restrict cell behaviors by limiting the degrees of freedom in which morphogenic phenomena can occur. Additionally, current tools to interrogate gene function, such as genetic knockouts or siRNA (*Boettcher and McManus, 2015*), cannot selectively perturb gene expression of subpopulations of cells in situ, which is required to generate controlled asymmetry analogous to embryonic morphogenesis.

Several of these limitations can be addressed with inducible CRISPR interference (CRISPRi) systems in mammalian cells (*Larson et al., 2013*; *Mandegar et al., 2016*). CRISPRi silencing enables temporal regulation over knockdowns (KD) of specific genetic targets with limited off-target effects. Temporal KD constraints enable the development of precisely-controlled engineered biological systems that can induce well-defined genetic perturbation at explicit times and within defined populations of cells to mimic developmental symmetry-breaking events.

Morphogenic asymmetries arise from reorganization of cells due to local changes in mechanical tissue stiffness and cell adhesions that facilitate physical organization of developing embryos (*Krieg et al., 2008*; *Maître et al., 2012*). Mechanical rearrangement is necessary for many aspects of morphogenesis, including cell polarity, collective movement, multicellular organization, and organ size regulation (*Arboleda-Estudillo et al., 2010*; *Maître, 2017*). Differential adhesion (*Foty and Steinberg, 2004*; *Foty and Steinberg, 2005*) and cortical tension (*Van Essen and Essen, 1997*;

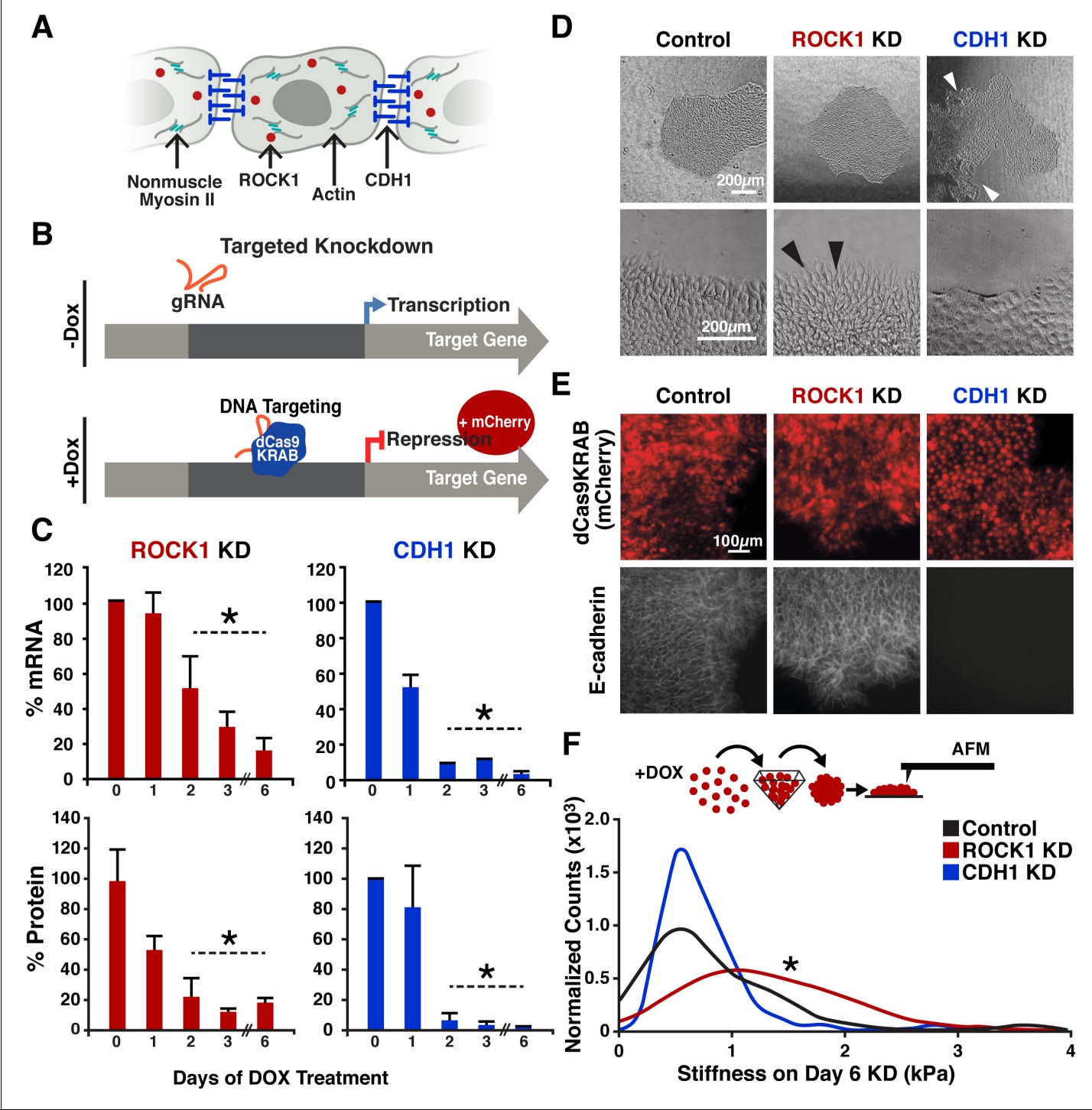

**Figure 1.** CRISPRi of ROCK1 and CDH1 modulate physical properties of the cell. (A) Schematic of ROCK1 and CDH1 within a cell. CDH1 is a trans-membrane adhesion molecule that locates to the borders of cells and ROCK1 is a cytoplasmic kinase that acts upon non-muscle myosin II. (B) Schematic of the CRISPRi system. Doxycycline addition to the hiPSC culture media leads to the expression of mCherry and dCas9-KRAB to induce knockdown of target gene. (C) qPCR and western blot quantification of knockdown timing; knockdown of both mRNA and protein were achieved by day three of DOX treatment when compared to untreated hiPSCs (p<0.05, n = 3, data represent mean ± SD). (D) Brightfield imaging of knockdown hiPSCs indicated morphological differences in colony shape (white arrows) and cell extensions (black arrows) at colony borders. (E) Live reporter fluorescence for dCas9-KRAB expression (red) and immunostaining for CDH1 (gray) demonstrated loss of CDH1 in induced CDH1 CRISPRi hiPSCs, but maintenance of CDH1 contacts in the off-target control and ROCK1 KD hiPSCs. (F) Atomic force microscopy (AFM) of knockdown populations exhibited

*Figure 1 continued on next page*

*Figure 1 continued*

a twofold increase in Young's elastic modulus of ROCK1 knockdown cells compared to control and CDH1 knockdown cells (p<0.05, n = 36, 65, 72 force points for Control, ROCK1 KD, and CDH1 KD, respectively, area under curve = 1).

DOI: https://doi.org/10.7554/eLife.36045.003

The following figure supplements are available for figure 1:

**Figure supplement 1.** Protein KD time course for ROCK1 and CDH1.

DOI: https://doi.org/10.7554/eLife.36045.004

**Figure supplement 2.** Morphology of ROCK1 KD hiPSCs.

DOI: https://doi.org/10.7554/eLife.36045.005

**Figure supplement 3.** Maintenance of tight junctions in KD hiPSCs.

DOI: https://doi.org/10.7554/eLife.36045.006

**Figure supplement 4.** Maintenance of normal hiPSC karyotypes.

DOI: https://doi.org/10.7554/eLife.36045.007

*Krieg et al., 2008*) are critical determinants of mechanically-driven cell sorting, in which both processes are known to contribute to tissue organization (*Lecuit and Lenne, 2007*). In cortical tension-dominated sorting, variable actin cytoskeleton-generated cortex tension stimulates sorting of individual cells, whereas differential adhesion sorting promotes segregation of cell populations due to intercellular homophilic adhesions.

Rho-associated coiled-coil containing protein kinase (ROCK1) and E-cadherin (CDH1) are interesting orthogonal gene targets to interrogate hPSC population organization by altering the intrinsic mechanics of distinct cell populations. ROCK1 regulates actin-myosin dynamics (*Figure 1A*), which contribute to a cell's cortical tension (*Salbreux et al., 2012*). In addition, ROCK inhibition is often used in hPSC culture and has been implicated in pluripotency maintenance (*McBeath et al., 2004*; *Ohgushi et al., 2015*). Similarly, CDH1, a classic type I cadherin adhesion molecule, is widely associated with pluripotency and early morphogenesis (*Heasman et al., 1994*; *Przybyla et al., 2016*; *Ringwald et al., 1987*), and its down-regulation parallels the induction of patterning events via differential adhesion (*Figure 1A*).

In this study, we explored whether mechanical manipulation of human induced pluripotent stem cells (hiPSCs) sub-populations results in controllable cell driven self-organization into repeatable patterns. We employed an inducible CRISPR interference (CRISPRi) system in hiPSCs to silence key proteins that regulate cell adhesion and cortical tension. We genetically induced controlled symmetry-breaking events within hiPSC populations by creating mixed populations of hiPSCs with and without the CRISPRi system and then induced mosaic knockdown (KD). Mosaic KD was employed to interrogate how the creation of physical asymmetries in an otherwise homogeneous population leads to multicellular organization and pattern formation. We show that induction of mosaic KD of ROCK1 or CDH1 results in a 'bottom-up' cell-driven pattern formation of hiPSC colonies while preserving pluripotency.

## Results

### CRISPRi KD in human iPSCs modulates epithelial morphology

To establish an inducible CRISPRi KD of ROCK1 or CDH1, we used a doxycycline (DOX)-inducible CRISPRi hiPSC line (CRISPRi-Gen1C) (*Mandegar et al., 2016*) (*Figure 1B*). Guide RNA (gRNA) sequences designed to target the transcription start site of ROCK1 or CDH1 (*Supplementary file 1* - Table 1) were introduced into CRISPR-Gen1C hiPSCs and KD was induced by the addition of DOX (2 µM) into cell culture media. After 3 days of KD induction, ROCK1 mRNA levels were <30% of hiPSCs without DOX treatment, and CDH1 mRNA levels in CDH1 KD hiPSCs were <10% compared to untreated controls (*Figure 1C*). Protein KD followed a similar trend where KD populations compared to untreated controls resulted in <20% ROCK1 protein and <10% of CDH1 protein by day three of DOX treatment, and reduced protein levels were maintained through day six of CRISPRi induction (*Figure 1C*, *Figure 1—figure supplement 1*).

Both the ROCK1 KD cells and the CDH1 KD cells retained epithelial hiPSC morphologies without single cell migration away from the colonies (*Figure 1D*). However, CDH1 KD hiPSCs displayed

irregular colony shapes that maintained smooth peripheral edges, but contained regions lacking cells within colonies (*Figure 1D*). Conversely, ROCK1 KD hiPSCs displayed round colony shapes (similar to wildtype hiPSCs) but individual cells along the border of ROCK1 KD colonies extended protrusions out away from the colony (*Figure 1D*). As expected, hiPSCs treated with the small-molecule ROCK inhibitor Y-27632 yielded a similar morphology to the ROCK1 KD hiPSCs with extended cell protrusions at the colony borders (*Figure 1—figure supplement 2*).

To further confirm the physical effects of knocking down CDH1 or ROCK1 selectively in hiPSCs, we performed immunofluorescent (IF) staining of CDH1 expression. After 5 days of DOX treatment, CDH1 KD hiPSCs exhibited a complete loss of CDH1 expression, as expected, whereas the ROCK1 KD hiPSCs and the control hiPSCs (with off-target CRISPRi guide) maintained robust expression of CDH1 along the plasma membrane (*Figure 1E*). To interrogate cell cortical tension, the contact angles between cells were measured based on IF of zona occluden-1 (ZO1), a protein associated with tight junctions (*Figure 1—figure supplement 3A*). Contact angles were not statistically different in either the ROCK1 KD or CDH1 KD cells compared to the control, but all populations displayed a subtle reduction in mean contact angle with DOX addition that was not significantly different between any of the groups (*Figure 1—figure supplement 3B*). However, when direct measurements of hiPSC elasticity were taken using atomic force microscopy after 6 days of KD, ROCK1 KD cells displayed a twofold higher cortical stiffness than the control and CDH1 KD populations, whereas the latter groups did not differ from one another (*Figure 1F*). Therefore, CRISPRi silencing of targeted genes associated with cellular mechanical properties resulted in distinct physical differences between the otherwise similar cell populations.

## Mosaic CRISPRi silencing results in multicellular organization

To examine whether mosaic KD of a single molecule impacted hiPSC organization, ROCK1- or CDH1-CRISPRi populations were pretreated with DOX for 5 days and mixed with isogenic wildtype hiPSCs that constitutively expressed GFP (WT-GFP) at a 1:3 ratio. Forced aggregation of ROCK1 KD: WT-GFP hiPSCs or CDH1 KD: WT-GFP hiPSCs and subsequent re-plating were used to create individual colonies of randomly mixed ROCK1 KD hiPSCs or CDH1 KD hiPSCs with the WT-GFP cells (*Figure 2A*). After 5 days in mixed culture, ROCK1 KD cells sorted radially from the WT-GFP cells, clustering primarily at the edges of the colonies (*Figure 2B,C*). However, separation of the ROCK1 KD cells did not result in distinct smooth borders between the WT-GFP and ROCK1 KD hiPSC populations. In contrast, CDH1 KD cells robustly separated from the GFP-WT population, forming sharp boundaries between populations irrespective of their spatial location within the colony (*Figure 2B, C*).

To determine whether pattern emergence was impacted by the relative proportion of mosaic KD within a colony, KD cells were mixed with control CRISPRi hiPSCs lacking any gRNA or fluorescent protein at varying cell ratios of 1:1, 1:3, and 3:1. Clustering of ROCK1 KD cells was less apparent as the proportion of ROCK1 KD cells within a colony increased. In fact, increasing ROCK1 KD hiPSCs to 75% of the colony resulted in the entire colony morphology displaying characteristics of a pure ROCK1 KD colony (*Figure 2—figure supplement 1*). On the other hand, the CDH1 CRISPRi cells separated from the colorless CRISPRi population, irrespective of cell ratio, indicating that the spatial organization occurred regardless of relative population size within a hiPSC colony. The ability of both the ROCK1 and CDH1 CRISPRi KD populations to physically partition from otherwise identical CRISPRi-engineered hiPSCs that lacked a gRNA confirms that the production of dCas9-KRAB is not responsible for the previously observed pattern formation when CRISPRi KD hiPSCs were mixed with the WT-GFP cells, but rather that the segregation is a direct result of KD of the target gene (*Figure 2—figure supplement 1*).

Based on the sorting behaviors of ROCK1 KD: WT-GFP and CDH1 KD: WT-GFP colonies when the KD of ROCK1 or CDH1 was present at the time of mixing, we next examined whether induction of mosaic KD after mixing resulted in similar sorting patterns as previously observed. This scenario more accurately models the onset of initial symmetry-breaking events among homogeneous pluripotent cells during embryonic development. Non-induced CRISPRi populations were mixed with WT-GFP hiPSCs (1:3 ratio), re-plated, and then treated with DOX to induce KD (*Figure 2E*). ROCK1 KD post-mixing within mosaic colonies did not result in noticeable radial segregation of ROCK1 KD hiPSCs from WT-GFP hiPSCs (*Figure 2F–H*), as observed for pre-mixed colonies. Instead, the post-mixing ROCK1 mosaic KD colonies exhibited greater vertical stacking of ROCK1 KD cells and WT-

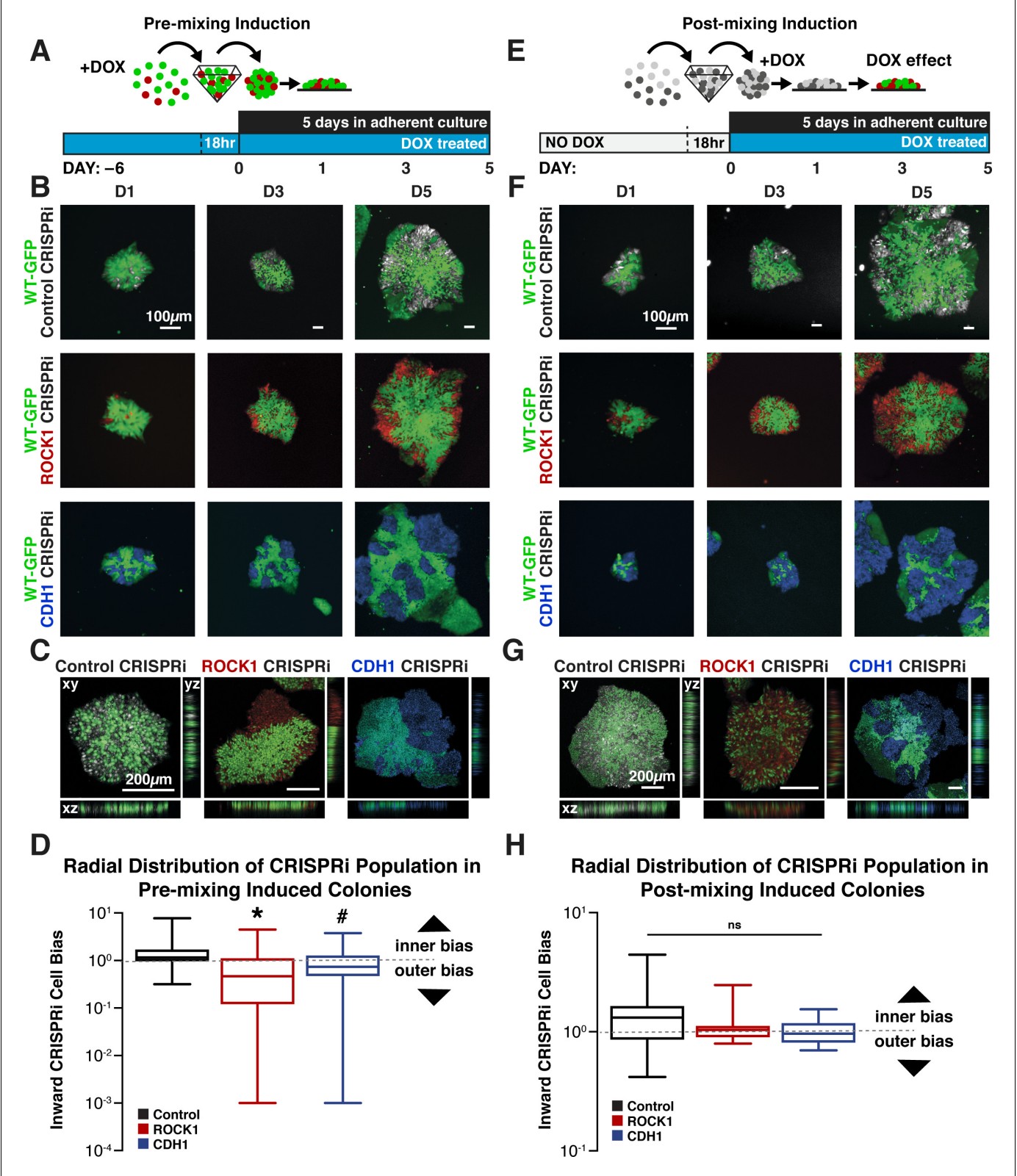

**Figure 2.** Cell-autonomous pattern emergence in mixed population colonies. (A) Schematic of experimental timeline. WT-GFP and ROCK1- or CDH1-CRISPRi hiPSCs were pretreated with doxycycline for 6 days before aggregation in pyramidal microwells and re-plating as mixed colonies. (B) Live cell imaging of pattern emergence over time from mixing colonies. Control populations remain mixed, ROCK1 KD hiPSCs cluster radially at borders of colonies, and CDH1 KD populations sort themselves from WT-GFP hiPSCs regardless of location within colony. (C) Confocal microscopy of patterned

*Figure 2 continued on next page*

*Figure 2 continued*

colonies of hiPSCs with KD induction prior to mixing. (**D**) Quantification of the radial distribution of KD cells in pre-induced mixed colonies. The ratio of inner cell area to outer cell area normalized to total cell area is displayed (n = 25,* and # indicate significance, p<0.05). (**E**) Schematic of experimental timeline for WT-GFP and ROCK1- or CDH1-CRISPRi hiPSCs treated with doxycycline upon re-plating as mixed colonies. (**F**) Live cell imaging of pattern emergence in post-mixing induction colonies, where CRISPRi KD is induced after cell population mixing. (**G**) Confocal microscopy of patterned hiPSC colonies with KD induction upon mixing populations, where ROCK KD cells stack vertically with WT-GFP hiPSCs. (**H**) Quantification of the radial distribution of KD cells in post-induced mixed colonies. The ratio of inner cell area to outer cell area normalized to total cell area is displayed (n = 20).
DOI: https://doi.org/10.7554/eLife.36045.008
The following figure supplements are available for figure 2:

**Figure supplement 1.** Pattern emergence of variable ratios of mixed populations.
DOI: https://doi.org/10.7554/eLife.36045.009
**Figure supplement 2.** Cell population change over time.
DOI: https://doi.org/10.7554/eLife.36045.010

GFP cells in the z-plane of the mixed colonies (*Figure 2G*), whereas the pre-induced mixed colonies

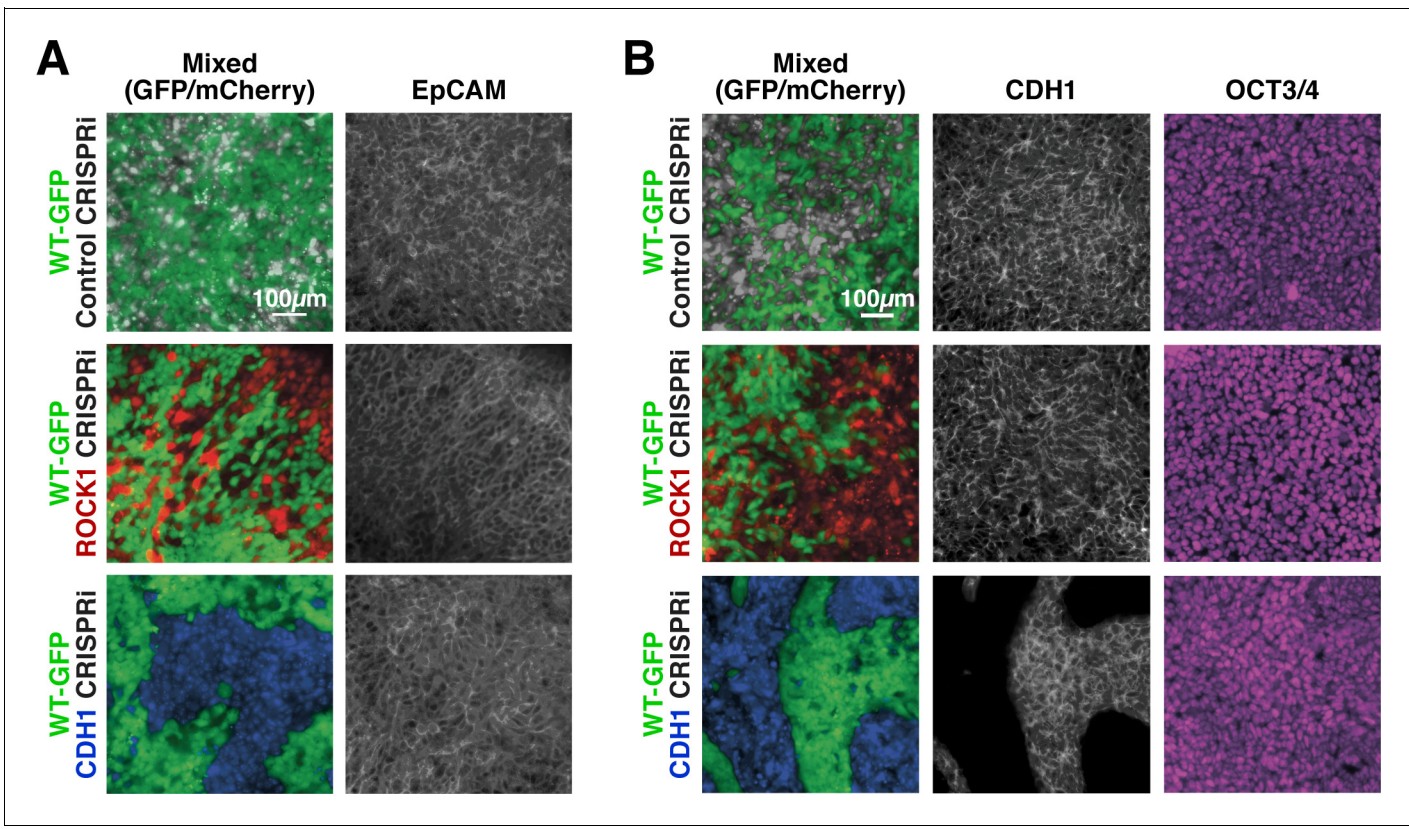

**Figure 3.** Maintenance of nuclear pluripotency markers and epithelial phenotype. (**A**) Immunostaining of EpCAM for mixed colonies displayed relatively uniform expression regardless of KD. (**B**) Immunostaining for E-cadherin (CDH1) and OCT3/4 in patterned hiPSC colonies demonstrating nuclear localized OCT3/4 throughout the mixed populations.
DOI: https://doi.org/10.7554/eLife.36045.011
The following figure supplements are available for figure 3:

**Figure supplement 1.** Maintenance of cell junction localized β-catenin in KD cells.
DOI: https://doi.org/10.7554/eLife.36045.012
**Figure supplement 2.** Pluripotent and early germ layer gene expression in KD cells.
DOI: https://doi.org/10.7554/eLife.36045.013

remained segregated primarily in a 2D planar manner (*Figure 2C*). In contrast, the mosaic silencing of CDH1 post-mixing maintained robust segregation of the CDH1 KD cells from the WT-GFP hiPSCs, although the borders between cell populations lacking CDH1 contacts and neighboring WT-GFP cells were somewhat less distinct than the pre-induced CDH1 KD: WT-GFP mixed colonies. Overall, the inducible CRISPRi mixed colonies displayed the ability to mimic several different patterns of intrinsic symmetry-breaking events that resulted in distinct cell sorting and multicellular pattern formation.

In addition to the changes in organization within colonies, significant changes were observed in the final cell ratios starting from an original seeding density of 3:1 WT to CRISPRi cells. The proportion of CRISPRi cells increased within the mixed colonies over time (*Figure 2—figure supplement 2A*). To determine if the accelerated growth was in response to mixing with a WT population, EdU incorporation was analyzed in pure CRISPRi and WT populations. Over 5 days of mixed culture, the WT-GFP cells displayed approximately 50% reduced DNA synthesis compared to the CRISPRi lines independent of DOX treatment (*Figure 2—figure supplement 2B*), however cell replication rate did not account for pattern formation as the CRISPRi control mixed colonies did not display evidence of any patterns.

## Mosaic hiPSC colonies retain a pluripotent phenotype

Colony morphology and expression of epithelial markers, such as epithelial cell adhesion molecule (EpCAM), were examined to determine if the cells that lost CDH1 expression segregated from their WT-GFP neighbors due to delamination, or loss of the epithelial phenotype characteristic of hiPSCs. ROCK1 KD: WT-GFP and CDH1 KD: WT-GFP colonies maintained an epithelial morphology throughout 6 days of CRISPRi silencing (*Figure 3A*) with no observed migration by CRISPRi-modulated cells away from the colonies. Both ROCK1 KD and CDH1 KD hiPSCs within mixed colonies expressed EpCAM at cell-cell boundaries after 6 days of CRISPRi induction despite changes in cortical tension or intercellular adhesion due to loss of ROCK1 or CDH1, respectively (*Figure 3A*). Furthermore, ROCK1/CDH1 KD hiPSCs displayed cell junction-localized β-catenin in pure colonies after 6 days of CRISPRi induction, suggesting maintenance of adherens junctions and epithelial colonies (*Figure 3— figure supplement 1A*).

Since the decrease of CDH1 is commonly associated with loss of pluripotency in PSCs, the expression and localization of the common pluripotency transcription factors, OCT3/4 and SOX2, were examined. Both proteins maintained strong nuclear expression in pure ROCK1 KD or CDH1 KD colonies after 6 days of KD induction (*Figure 3—figure supplement 2A*). Moreover, despite the physical segregation of cells induced by KD in mixed populations, no pattern could be observed based on pluripotency marker expression (*Figure 3B*). Furthermore, the abundance of OCT3/4 and SOX2 transcripts was unchanged in pure colonies of CDH1 KD cells and not significantly different, though variable, in pure colonies of ROCK1 KD hiPSCs (*Figure 3—figure supplement 2B*). However because the transcription factors SOX2 and OCT4 are retained by cells for a period of time during the process of differentiation, genes associated with the primitive streak (Brachyury [BRA]) and the neural crest (SOX9) were interrogated in either ROCK1 or CDH1 KD cells over 6 days (*Figure 3—figure supplement 2C*). Both BRA and SOX9 were significantly increased on day three of KD in ROCK1 KD cells, however at day six the gene expression returned to levels comparable to day zero before ROCK1 KD. Although the CDH1 KD cells did not display any significant trends, the standard deviation of gene expression varied as much as three times greater than that of the ROCK1 KD cells. The high variation between biological replicates potentially indicates that silencing of CDH1 induces a large variability in the gene regulation of BRA and SOX9 and could indicate that the cells experience a transient fluctuation in the pluripotency state. However, these results indicate that the loss of ROCK1 or CDH1 is not sufficient to disrupt the pluripotent gene regulatory network and induce an immediate exit from the pluripotent state.

## Mosaic hiPSC patterns display transient gene expression changes in coordination with emergence of patterns

Since pluripotency markers were maintained irrespective of mosaic patterning, gene expression changes in pluripotency markers (SOX2, NANOG), mesendoderm markers (SOX17, BRA) and ectoderm markers (PAX6, SOX9) were examined during the course of mosaic patterning at days 1, 3 and

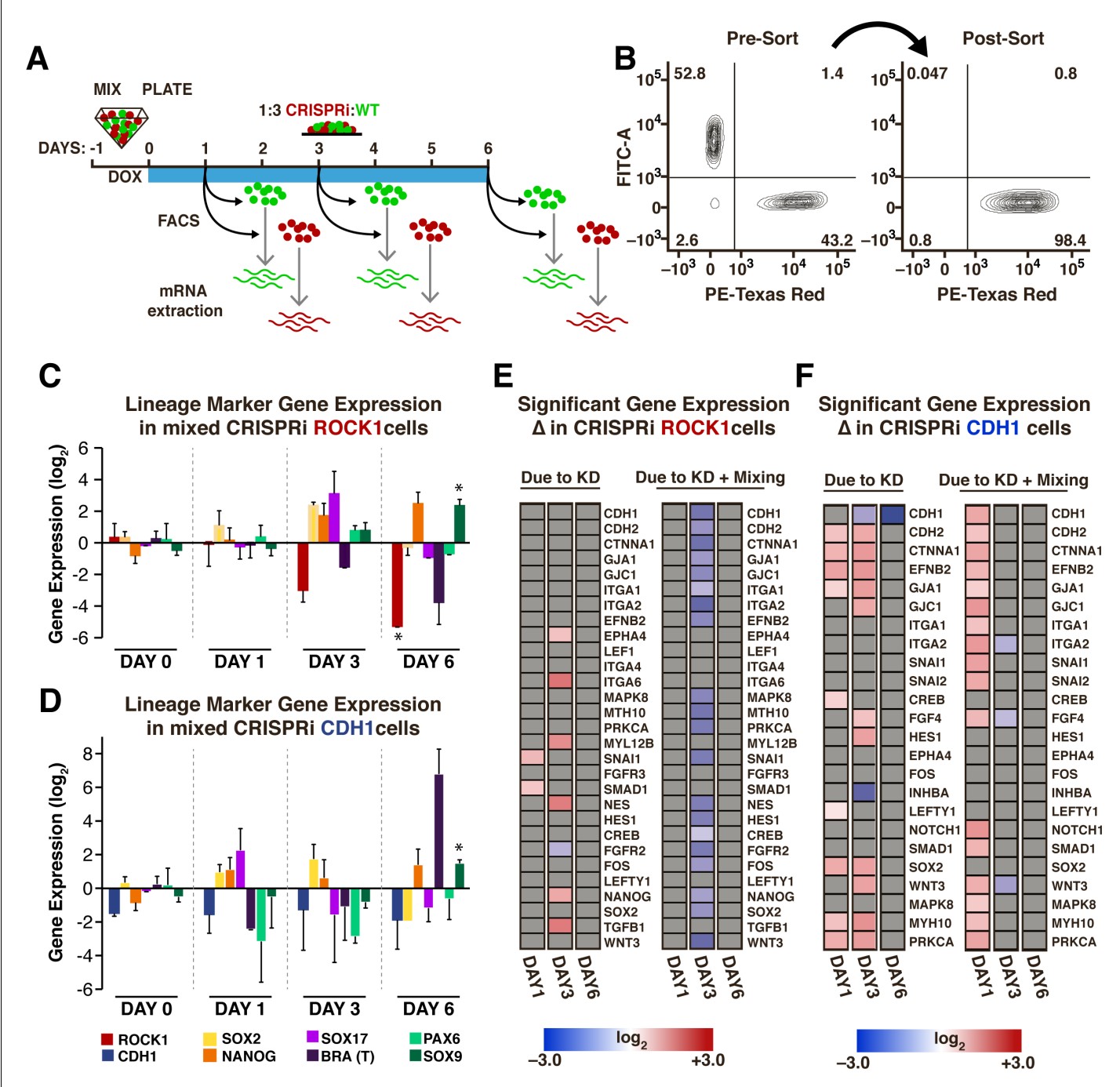

**Figure 4.** Transient gene expression changes in mixed populations. (A) Schematic of experimental timeline; WT-GFP and ROCK1- or CDH1-CRISPRi hiPSCs were mixed and re-plated prior to KD induction. Different cell populations were isolated by FACS for mRNA extraction on days 1, 3, and 6 after KD induction. (n = 3 per condition). (B) Representative scatter plot of a FACS-sorted population of mCherry +cells (indicating KD induction) with >98% purity. (C,D) Plots of specific mRNA expression changes at days 1, 3, and 6 in KD cell populations that have been mixed with WT. (* and # indicate significance, p<0.05). (E,F) Heat maps display fold change expression of genes found to display significant changes in ROCK1 or CDH1 KD cells mixed with WT-GFP hiPSCs when compared to time-matched, off-target control hiPSCs. Grey color indicates non-significance. Significance (p<0.05, n = 3) was determined using a one-way analysis of variance (ANOVA) followed by post-hoc pairwise comparisons by Tukey's tests to determine the effect of mixing populations, the effect of solely KD, and the effect of KD within a mixed population.

DOI: https://doi.org/10.7554/eLife.36045.014

The following figure supplements are available for figure 4:

**Figure supplement 1.** Gene expression changes in WT-GFP cells mixed with CRISPRi cells.

*Figure 4 continued on next page*

*Figure 4 continued*

DOI: https://doi.org/10.7554/eLife.36045.015

**Figure supplement 2.** Germ lineage differentiations of mixed colonies.

DOI: https://doi.org/10.7554/eLife.36045.016

6 after KD induction (*Figure 4A*). To take into account potential gene expression changes that result from mixing hiPSC lines, un-induced mixed populations and un-induced pure populations were analyzed as controls. BRA did not change significantly with induction of ROCK1 KD or CDH1 KD in a mixed population, however SOX9 increased on day six of KD in both ROCK1 KD and CDH1 KD cells (*Figure 4C,D*). Interestingly, similar to pure populations, there was a large variance in gene expression between biological replicates, often displaying more than a onefold change difference in gene expression between biological replicates in mixed colonies.

To assess whether gene expression changes were unique to the induction of symmetry breaking events in mixed populations or simply a result of gene KD, a curated set of genes involved in pluripotent stem cell signaling, early lineage fate transitions, and regulation of physical cell properties (*Supplementary file 1* - Table 2) was examined in both pure KD populations and mixed KD populations. An ANOVA analysis was used to examine gene expression changes that could be attributed to mixing two different cell types (mixed populations without KD), to KD of ROCK1 or CDH1 in a pure population, and to mosaic KD or KD in the presence of a WT neighbor (*Figure 4E,F*). Overall, few changes in gene expression resulted from mixing un-induced CRISPRi populations with WT-GFP (*Figure 4—figure supplement 1A*), and therefore all subsequent data were normalized to pure un-induced populations and then to mixed un-induced populations to minimize false positives that resulted from mixing of cell lines without induction of KD.

In ROCK1 KD cells mixed with WT, the gene expression changes on day one that could be attributed solely to gene KD (*Figure 4E*, left column) were associated with primitive streak formation (Snail (SNAI1), SMAD1). On day three, an upregulation of adhesion molecules (EPHA4, ITGA4) was observed, as well as NANOG, Nestin (NES), and TGFβ upregulation and FGFR2 down-regulation. Interestingly, a large number of changes in gene expression were specific to the mosaic induction of ROCK1 KD (*Figure 4E*, right column), for example the down-regulation of both cell-cell adhesions as well as cell-ECM adhesions. Additionally, genes that were upregulated in the pure KD context were down-regulated in the mosaic KD context, such as SNAI1, NES and NANOG. However, at day six of mosaic KD no significant changes persisted in the examined panel of genes (*Figure 4E*). CDH1 KD caused an upregulation in genes associated with cell-cell adhesion on day one that was exacerbated in a mosaic CDH1 KD (*Figure 4F*). Interestingly, both Wnt3 and down-stream Wnt targets, such as SNAI1 and SNAI2, were significantly increased specifically in mosaic KDs on day one of KD. Similar to the transient wave of gene expression changes observed in ROCK1 KD cells, mosaic CDH1 KD did not exhibit any observed significant changes on day six of KD except for CDH1 (*Figure 4F*). These results were consistent with our previous observation of the maintenance of pluripotency in the ROCK1 KD: WT-GFP and CDH1 KD: WT-GFP colonies (*Figure 3*). Furthermore, the recovery of homeostatic gene expression profiles closely followed the dynamics of distinct pattern establishment in the mixed populations.

In addition to examining the KD cells, we examined the gene expression profiles of the neighboring WT cells that constituted the majority of cells in each colony. On day six of KD induction, the WT-GFP cells that were mixed with CDH1 KD hiPSCs had gene expression patterns that resembled the WT-GFP cells mixed with the control CRISPRi populations, whereas the WT-GFP cells mixed with ROCK1 KD hiPSCs exhibited a different expression profile. Interestingly, the WT-GFP cells mixed with ROCK1 CRISPRi hiPSCs demonstrated changes in genes associated with cell sorting and movement, such as ephrins and integrins, and up-regulation in myosin proteins (MYH9, MYH10) (*Figure 4—figure supplement 1B,C*). Overall, the changes in the WT-GFP hiPSC gene expression suggests that targeted manipulation of gene expression in an emerging sub-population can exert non-cell– autonomous effects on the opposing population and may be influenced by the respective multicellular organization of the two populations.

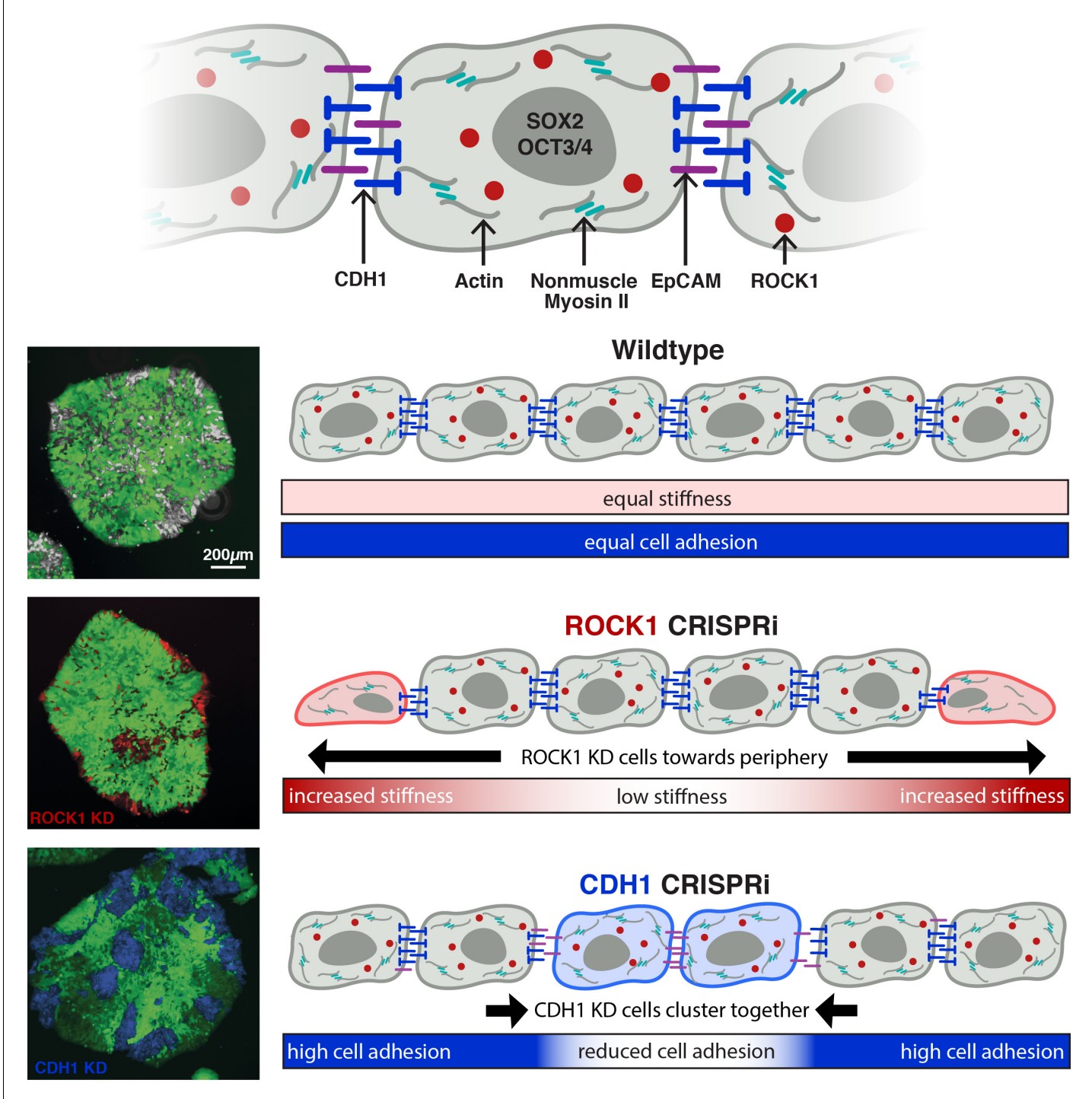

**Figure 5.** Inducible pattern emergence through the KD of molecules that affect hiPSC physical properties. (**A**) Schematic of working model of sub-population manipulation where controlled changes in cellular stiffness or cellular adhesion result in specific colony pattern formation. With mosaic KD, ROCK1 produces continuous radial separation of KD cells from WT, whereas CDH1 displays discrete islands of KD cells within the WT population.
DOI: https://doi.org/10.7554/eLife.36045.017

## Mixed populations direct germ lineage emergence

Controllable induction of two distinct populations of hPSC offers the potential for co-emergence of multiple differentiated cell populations in a predictable manner. To examine how mosaic patterning

of hPSC could direct co-emergence of differentiated progeny, two independent differentiation protocols were performed to direct the hiPSCs to either an ectodermal or mesendodermal fate (*Figure 4—figure supplement 2A*). The proportion of PAX6 +cells, indicating neuro-ectoderm lineage, or eomesodermin (EOMES) +cells, indicating a mesendoderm lineage, in the WT and CRISPRi mixed populations were examined. Although the ROCK1 KD population did not display a significant difference in PAX6 +or EOMES +cells relative to the WT cells (*Figure 4—figure supplement 2B-D*), the CDH1 KD population yielded fewer PAX6 +cells with the ectoderm directed differentiation and increased EOMES +cells in the mesendoderm-directed differentiation (*Figure 4—figure supplement 2B-D*). Overall, these studies demonstrate the potent ability to direct multicellular organization of hPSCs prior to the acquisition of differentiated cell fate.

## Discussion

In this study we examined the effect of inducing specific genetic KD in subpopulations of hiPSCs within an otherwise homogeneous population of pluripotent cells. Historically, small-molecule chemical inhibitors, antibodies, and homogeneous genetic knockouts are often used to interrogate the molecular mechanisms involved in morphogenesis (*Lecuit and Lenne, 2007*; *McBeath et al., 2004*; *Salbreux et al., 2012*). However, these methods can't selectively discriminate between different cells, or they fail to address how the emergence of heterotypic interactions affects multicellular organization. Here, we report that silencing of target genes by CRISPRi within only subpopulations of cells provides multiple avenues to genetically control the emergence of asymmetric cell phenotypes and development of multicellular patterns. Specifically, we demonstrate that mosaic KD of target genes ROCK1 or CDH1 result in distinct patterning events wherein cell-driven segregation dictates colony organization without loss of pluripotency (*Figure 5*).

ROCK1 regulates actin-myosin contraction (*McBeath et al., 2004*) and facilitates expansion of PSCs (*Ohgushi et al., 2015*; *Park et al., 2015*), and its acute inhibition by small molecules leads to a 'relaxed' cell phenotype with decreased stiffness (*Kinney et al., 2014*; *Lee et al., 2006*). However, we found that prolonged silencing of ROCK1 in hiPSCs (6 days) resulted in cells that were twofold stiffer than either the CDH1 KD cells or the control CRISPRi cells. The increased cortical stiffness of ROCK1 KD hiPSCs could be due to the difference between the inhibition of an existing protein and KD of the gene. A small molecule inhibitor prevents the function of already existing proteins so that a small amount of functioning protein may escape the inhibitor's influence. In contrast, CRISPRi only needs to target the ROCK1 gene loci at two alleles to completely abolish protein transcription, thus highlighting the strength of genetic perturbation. Additionally, ROCK inhibition is often used as a transient perturbation (24 hr), whereas long-term KD of ROCK1 (6 days) may induce compensatory effects within the cells that are responsible for the somewhat surprising results. Long-term ROCK1 KD compensation is a likely partial explanation why KD of ROCK1 prior to mixing resulted in radially partitioned populations, but post-mixing KD resulted in less segregated populations. The increased stacking of the ROCK1 KD population upon induction of knockdown after mixing with WT cells could also reflect the difference between short-term and long-term ROCK1 KD and potential compensation effects. For example, the 3D accumulation of ROCK1 KD cells as cytoskeletal contraction is impaired may be a result from compensatory contraction by the surrounding WT cells, resulting in multilayer colonies.

The emergence of autonomous cell patterning and cell sorting is often coincident with the onset of differentiation, and CDH1 in particular regulates morphogenesis in a diverse range of species (*Burdsal et al., 1993*; *Li et al., 2010*). Historically, CDH1 is often associated with pluripotency in mouse embryonic stem cells (*Li et al., 2012*; *Soncin and Ward, 2011*). However, while CDH1 is commonly expressed by pluripotent cells and CDH1 can replace OCT3/4 during fibroblast reprogramming to pluripotency (*Redmer et al., 2011*), CDH1 is not essential to pluripotency (*Larue et al., 1996*; *Soncin et al., 2009*; *Ying et al., 2008*). Our results reinforce these latter observations by demonstrating that CDH1 KD in hiPSCs does not adversely affect the expression of pluripotency markers nor lead to a loss of epithelial phenotype. The maintenance of pluripotent state indicates that KD of CDH1 alone is not sufficient to induce differentiation, but other factors such as sufficient cell density and local intercellular signaling preserve pluripotent colony integrity and buffer against changes in cell state.

Furthermore, the observed maintenance of pluripotency and preferential differentiation to mesendoderm is consistent with recent studies revealing that changes in human CDH1 adhesions coordinate with *in vitro* human stem cell lineage decisions rather than pluripotency maintenance (*Przybyla et al., 2016*). Changes in CDH1 influencing lineage fate decisions may explain the transient gene expression changes that we observed with the induction of KD in mixed colonies, where the loss of CDH1 potentially primes the cells to respond to a signal for differentiation, and without such a signal the cells return to a ground state of pluripotency. Similar priming has been described in the context of cell-matrix adhesion where differentiation in response to TGFβ signaling is primed by stiffness-dependent integrin signaling (*Allen et al., 2012*); a similar mechanism may explain the observed transient gene expression changes without loss of pluripotency in CDH1 KD hiPSCs.

The ability to manipulate distinct cell populations allows for robust modeling of human morphogenic events and, thus, an expanded understanding of human biology that can be exploited to develop physiologically realistic *in vitro* human tissue models. Cellular location within pluripotent colonies can be thought to parallel the effects seen in early developing blastocysts. A cell's location within the early embryo relays signals that dictate initial symmetry-breaking events, such as the decision to become trophectoderm instead of inner cell mass. Cells located within the center of an embryo maintain different adhesion contacts (*Stephenson et al., 2010*) and are subjected to higher tension generated by neighboring cells (*Samarage et al., 2015*), which then feed back into lineage fate decisions. For example, the Hippo pathway is controlled by a cell's position within the early blastocyst, where the outer cell layer has the ability to polarize and sequester the signaling molecule angiomotin away from adherens junctions, preventing the phosphorylation and activation that would occur in an internal cell that maintained cell-cell contacts on all sides (*Hirate et al., 2013*). Additionally *in vitro* micro-patterned PSC colonies have been reported to display spatially dependent germ layer patterning upon differentiation (*Etoc et al., 2016*; *Tewary et al., 2017*; *Warmflash et al., 2014*). The observed mosaic patterns of ROCK1 KD hiPSCs demonstrate that the spatial location within a colony can be affected by targeted gene KD without altering differentiation potential, and conversely, the mosaic CDH1 KD demonstrates dual control of spatial positioning and lineage potential. Therefore, the system described here enables interrogation of multicellular organization and morphogenic processes in parallel via manipulation of local multicellular domains through subpopulation organization and priming to specific lineage fates.

In addition to changes in multicellular organization as a result of ROCK1 or CDH1 KD, changes in gene expression also occurred in the WT population. In particular, the WT population displayed significant changes in several genes associated with adhesion and lineage fate. For example, GATA4 was down-regulated in WT cells in all mixed colonies. GATA4 is expressed by mesendoderm lineages (*Molkentin et al., 1997*; *Zorn and Wells, 2009*) and its down-regulation may affect the ability of WT cells to properly differentiate to mesendoderm. However, there were no significant differences in EOMES expression between the WT and CRISPRi cells when ROCK1 KD mixed colonies were directed toward a mesendodermal fate (*Figure 2—figure supplement 2*), potentially due to the strength of the small molecule CHIR, a GSK3 inhibitor, in inducing mesendoderm fate. Additionally, differences between the WT and ROCK1 KD populations may arise in longer differentiations, where the cells are allowed to mature beyond a progenitor stage. However, differences in maturation or cell type within a germ lineage may assist in the controlled co-emergence of multiple cell types that normally interact within a single tissue, such as parenchymal cells along with stromal populations.

Overall, this study capitalized on the ability of CRISPRi to temporally perturb specific molecular regulators of physical cell properties, such as adhesion and tension, that resulted in differing multicellular patterns. Moreover, CRISPRi additionally offers the flexibility to target any gene of interest and timing of KD (*Gordon et al., 2016*; *Mandegar et al., 2016*), allowing for the creation of dynamic patterns through transient genetic KD that could be used to pre-pattern PSC colonies in various types of multicellular geometries before differentiation. Additionally, the ability to induce molecular asymmetry can also be applied to co-differentiation, where the temporal induction of specific heterotypic interactions, such as the presentation of a ligand or receptor, can give rise to the coordinated emergence of two (or more) cell types under the same culture conditions. In addition, mosaic induction of KD can be used to examine how signals propagate between cells, for example, interrogating how the networks between cells created by either mechanical (adhesions) or chemical gradients (gap junctions) affect lineage fate decisions. Furthermore, the predictable patterning events and potential for control over co-emergence that we establish in this study could aid the

eventual control over morphogenic events in organoid systems. Organoids require coordinated heterotypic interactions in a 3D environment in order to self-organize (*Bredenoord et al., 2017*; *Sasai, 2013*); the ability to precisely predict and control the organization of multiple cell types in parallel would significantly improve the reproducibility and robustness of *in vitro* tissue modeling. Ultimately, this study identifies a novel strategy to direct the emergence of heterotypic cell populations to control multicellular organization in pluripotent stem cells, and subsequently facilitates the creation of robust models of morphogenesis necessary for the mechanistic study of human developmental tissue patterning and formation.

# Materials and methods

## Key resources table

| Reagent type (species) or resource | Designation | Source or reference | Identifiers | Additional information |
|---|---|---|---|---|
| Cell line (*H. sapien*, male) | WT-GFP | this paper | | hiPSC line containing constitutative GFP in AAVS1 locus |
| Cell line (*H. sapien*, male) | CRISPRi no guide | *Mandegar et al., 2016*; *DOI 10.1016/j.stem.2016.01.022* | | hiPSC line containing DOX inducible dCas9KRAB |
| Cell line (*H. sapien*, male) | CRISPRi control | *Mandegar et al., 2016*; *DOI 10.1016/j.stem.2016.01.022* | | hiPSC line containing DOX inducible dCas9KRAB and gRNA to KCNH2 |
| Cell line (*H. sapien*, male) | CRISPRi ROCK1 | *Mandegar et al., 2016*; *DOI 10.1016/j.stem.2016.01.022* | | hiPSC line containing DOX inducible dCas9KRAB and gRNA to ROCK1 |
| Cell line (*H. sapien*, male) | CRISPRi CDH1 | this paper | | hiPSC line containing DOX inducible dCas9KRAB and gRNA to CDH1 |
| Antibody | mouse anti-ROCK1 | AbCAM | ab58305 | (1:200) |
| Antibody | mouse anti-CDH1 | AbCAM | ab1416 | (1:200) |
| Antibody | goat anti-GAPDH | Invitrogen | PA1-9046 | (1:10000) |
| Antibody | goat anti-OCT3/4 | SantaCruz | sc8629 | (1:400) |
| Antibody | mouse anti-SOX2 | AbCAM | ab7935 | (1:400) |
| Antibody | mouse anti-Zo1 | Life Technologies | lifetech 339100 | (1:400) |
| Antibody | rabbit anti-NANOG | AbCAM | ab21624 | (1:300) |
| Antibody | mouse anti-Bcatenin | BD Biosciences | BD610154 | (1:200) |
| Antibody | mous anti-EpCAM | Millipore | MAB4444 | (1:200) |
| Antibody | Alexa 488- or 647 - secondaries | Life Technologies | | (1:500) |
| Other | Hoescht stain | Life Technologies | | (1:10000) |

*Continued on next page*

*Continued*

| Reagent type (species) or resource | Designation | Source or reference | Identifiers | Additional information |
|---|---|---|---|---|
| Recombinant DNA reagent | gRNA-CKB (plasmid) | *Mandegar et al., 2016*; *DOI 10.1016/j.stem.2016.01.022* | | vector containing gRNA and selection markers (blasticidin resistance, mKate fluorescence) |
| Software, algorithm | GraphPad Prism | GraphPad Prism (https://graphpad.com) | RRID:SCR_015807 | |
| Software, algorithm | ImageJ | ImageJ (http://imagej.nih.gov/ij/) | RRID:SCR_003070 | |
| Software, algorithm | Python, scikit image | *scikit-image contributors et al., 2014*: *DOI 10.7717/peerj.453* | | |
| Chemical compound, drug | mTeSR1 medium | STEMCELL Technologies | STEMCELL Technologies: 85850 | |
| Chemical compound, drug | Matrigel | Corning Life Sciences | Corning Life Sciences: 356231 | |
| Chemical compound, drug | Accutase | STEMCELL Technologies | STEMCELL Technologies: 7920 | |
| Chemical compound, drug | Blasticidin | ThermoFisher Scientific | ThermoFisher Scientific: R21001 | |
| Chemical compound, drug | Doxycycline | Sigma Aldrich | Sigma Aldrich: D9891 | |
| Chemical compound, drug | Y-27632 ROCK inhibitor | Selleckchem | Selleckchem: S1049 | |
| Chemical compound, drug | Puromycin | Sigma Aldrich | Sigma Aldrich: P8833 | |
| Chemical compound, drug | SB 435142 | Stemgent | Stemgent: 04-0010-05 | |
| Chemical compound, drug | LDN 193189 | Selleckchem | Selleckchem: S2618 | |
| Chemical compound, drug | CHIR 99021 | Selleckchem | Selleckchem: CT99021 | |
| Chemical compound, drug | TRIzol LS Reagent | ThermoFisher Scientific | ThermoFisher Scientific: 10296028 | |
| Sequenced-based reagent | gRNAs | This paper | | See *Supplementary file 1* |
| Sequenced-based reagent | RT-qPCR primers | This paper | | See *Supplementary file 1* |
| Commercial assay or kit | MycoAlert Mycoplasma Detection Kit | Lonza | Lonza: LT07218 | |
| Commercial assay or kit | Pierce BCA Protein Assay kit | ThermoFisher Scientific | ThermoFisher Scientific: 23250 | |
| Commercial assay or kit | RNeasy Mini Kit | QIAGEN | QIAGEN: 74106 | |
| Commercial assay or kit | iScript cDNA Synthesis kit | BIORAD | BIORAD: 1708891 | |
| Commercial assay or kit | Fast SYBR Green Master Mix | ThermoFisher Scientific | ThermoFisher Scientific: 4385612 | |
| Commercial assay or kit | Click-iT EdU Alexa 647 Imaging Kit | ThermoFisher Scientific | ThermoFisher Scientific: C10340 | |
| Commercial assay or kit | Direct-zol RNA MiniPrep Plus kit | ZYMO Research | ZYMO: R2061 | |

## Human iPSC line generation and culture

All work with hiPSC lines was approved by the University of California, San Francisco Human Gamete, Embryo and Stem Cell Research (GESCR) Committee. Human iPSC lines were derived from the WTC 11 line (Coriell Cat. # GM25256) where the species of origin was confirmed by a LINE assay. After genetic manipulation, all cell lines were karyotyped by Cell Line Genetics and were deemed karyotypically normal before proceeding with experiments (*Figure 1—figure supplement 4B*). All cell lines tested negative for mycoplasma using a MycoAlert Mycoplasma Detection Kit (Lonza).

Human iPSC lines were cultured in feeder-free media conditions on growth factor-reduced Matrigel (Corning Life Sciences) and fed daily with mTeSR™-1 medium (STEMCELL Technologies) (*Ludwig et al., 2006*). Accutase (STEMCELL Technologies) was used to dissociate hiPSCs to single cells during passaging. Cells were passaged at a seeding density of 12,000 cells per $cm^2$ and the small molecule Rho-associated coiled-coil kinase (ROCK) inhibitor Y-276932 (10 µM; Selleckchem) was added to the media upon passaging to promote survival (*Park et al., 2015*; *Watanabe et al., 2007*).

The generation of the ROCK1 CRISPRi line was previously created and described by *Mandegar et al., 2016*. For the generation of the CDH1 CRISPRi lines, five CRISPRi gRNAs were designed to bind within 150 bp of the TSS of CDH1 and cloned into the gRNA-CKB vector using BsmBI ligation following the previously described protocol (*Mandegar et al., 2016*) (*Supplementary file 1* -Table 1). gRNA expression vectors were nucleofected into the CRISPRi-Gen1C human hiPSC line from the Conklin Lab using the Human Stem Cell Nucleofector Kit 1 solution with the Amaxa nucleofector 2b device (Lonza). Nucleofected cells were seeded into 3 wells of a 6-well plate (~7400 cell/$cm^2$) in mTeSR™-1 media with Y-27632 (10 µM) for 2 days, and treated with blasticidin (ThermoFisher Scientific; 10 µg/ml) for a selection period of 7 days. Surviving colonies were pooled and passaged in mTeSR™-1 with blasticidin and Y-27632 for a single day then transitioned to mTeSR™-1 media only. Once stable polyclonal populations of CDH1 CRISPRi hiPSCs for each of the five guides were established, the cells were incubated with doxycycline (2 µM) for 96 hr. KD efficiency was evaluated by mRNA collection and subsequent qPCR, comparing levels of transcript with a time-matched control of the same line without CRISPRi induction. The CRISPRi CDH1 cell line with the guide producing the best KD was selected (gRNA −6).

To generate the WT-GFP line, 2 million WTC clone11 hiPSCs were nucleofected as previously described with the knock-in plasmid containing a CAG promoter-driven EGFP and AAVS1 TALEN pair vectors (*Figure 1—figure supplement 4A*). After cell recovery, puromycin (0.5 µg/ml) was added to the media for a selection period of 7 days. Individual stable EGFP-expressing colonies were picked using an EVOS FL microscope (Life Technologies) and transferred to individual wells of a 24-well plate in mTeSR media with Y-27632 (10 µM) and subsequently expanded into larger vessels.

## Generation of mixed colonies

Cell aggregates of ~100 cells were created using 400 × 400 µm PDMS microwell inserts in 24-well plates (~975 microwell per well) similar to previously published protocols (*Hookway et al., 2016*; *Ungrin et al., 2008*). Dissociated hiPSC cultures were resuspended in mTeSR™-1 supplemented with Y-27632(10 µM), mixed at proper ratios and concentration (100 cells/well), added to microwells, and centrifuged (200 rcf). After 18 hr of formation, 100 cell aggregates were transferred in mTeSR™-1 to Matrigel-coated 96-well plates (~15 aggregates/$cm^2$) and allowed to spread into 2D colonies.

## Western blot

Human iPSCs were washed with cold PBS, incubated for 10 min on ice in RIPA Buffer (Sigma-Aldrich), and supernatant collected. Three replicates were used for each condition. The supernatant protein content was determined using a Pierce BCA Protein Assay kit (Thermofisher Scientific) colorimetric reaction and quantified on a SpectraMax i3 Multi-Mode Platform (Molecular Devices). Subsequently, 20 µg of protein from each sample was resolved by SDS-PAGE, and transferred to a nitrocellulose membrane (Invitrogen). The membranes were incubated overnight at 4°C with primary antibodies: anti-ROCK1 (AbCAM 1:200), anti-CDH1 (AbCAM 1:200), anti-GAPDH, (Invitrogen 1:10,000), followed by incubation (30 min at room temperature) with infrared secondary antibodies:

IRDye 800CW and IRDye 680CW (LI-COR 1:13,000), and imaged on the Odyssey Fc Imaging System (LI-COR Biosciences). Protein levels were quantified using Image Studio Lite (LI-COR Biosciences).

## RNA isolation and qPCR

Total RNA isolation was performed using an RNeasy Mini Kit (QIAGEN) according to manufacturer's instructions and quantified with a Nanodrop 2000c Spectrometer (ThermoFisher Scientific). cDNA was synthesized by using an iScript cDNA Synthesis kit (BIORAD) and the reaction was run on a SimpliAmp thermal cycler (Life Technologies). To quantify individual genes, qPCR reactions were run on a StepOnePlus Real-Time PCR system (Applied Biosciences) and detected using Fast SYBR Green Master Mix (ThermoFisher Scientific). Relative gene expression was determined by normalizing to the housekeeping gene 18S rRNA, using the comparative threshold ($C_T$) method. Gene expression was displayed as fold change of each sample (ROCK1 CRISPRi or CDH1 CRISPRi) versus the off-target guide control (KCNH2 CRISPRi). The primers were designed using the NCBI Primer-BLAST website and are listed in *Supplementary file 1* -Table 2. Statistical analysis was conducted using a two-tailed unpaired *t*-test between any two groups (p<0.05, n = 3).

## Atomic force microscopy

All AFM indentations were performed using an MFP3D-BIO inverted optical atomic force microscope (Asylum Research) mounted on a Nikon TE2000-U inverted fluorescent microscope. Silicon nitride cantilevers were used with spring constants ranging from 0.04 to 0.06 N/m and borosilicate glass spherical tips 5 μm in diameter (Novascan Tech). Each cantilever was calibrated using the thermal oscillation method prior to each experiment. Samples were indented at 1 μm/s loading rate, with a maximum force of 4 nN. Force maps were typically obtained as a $6 \times 6$ raster series of indentations utilizing the FMAP function of the IGOR PRO build supplied by Asylum Research, for a total of 36 data points per area of interest measured every five microns. Two 5 micron by five micron areas of interest were sampled for each sample. The Hertz model was used to determine the elastic modulus of the sample at each point probed. Samples were assumed to be incompressible and a Poisson's ratio of 0.5 was used in the calculation of the Young's elastic modulus.

## Time-lapse imaging

Human iPSC colonies were imaged in 96-well plates (ibidi) on an inverted AxioObserver Z1 (Ziess) with an ORCA-Flash4.0 digital CMOS camera (Hamamatsu). Using ZenPro software, colony locations were mapped and a single colony was imaged daily for 6 days. To obtain time-lapse movies, a single colony was imaged over the course of 12 hr at a rate of one picture taken every 30 min.

## Immunofluorescence staining

Human iPSC colonies were fixed for 30 min in 4% paraformaldehyde (VWR) and washed 3X with PBS. Fixed colonies were permeabilized with 0.3% Trition X-100 (Sigma Aldrich) throughout blocking and antibody incubation steps. Samples were incubated in primary antibodies over night at 4°C, subsequently washed with PBS and incubated in secondary antibodies for an hour at room temperature. Primary antibodies used were: anti-OCT4 (SantaCruz 1:400), anti-SOX2 (AbCAM 1:400), anti-Zo1 (LifeTechnologies 1:400), NANOG (AbCAM 1:300), anti-β-catenin (BD Biosciences 1:200), anti-EpCAM (Millipore 1:200). All secondary antibodies were used at 1:1000 and purchased from Life Technologies.

## EdU incorporation

Pure populations of WT, CRISPRi KCNH2 (control), CRISPRi ROCK1, and CRISPRi CDH1 were treated with DOX (2 μM) for 5 consecutive days. Cultures were pulsed with EdU by supplementing Click-It EdU (10 μM) to the media for 6 hr. Cultures were then washed 3X with PBS and fixed with 4% paraformaldehyde (VWR) for 15 min and subsequently washed with PBS. Samples were permeabilized with 0.5% Triton-X 100 (Sigma Aldrich) in PBS for 20 min. Samples were then incubated with Click-It EdU detection kit as per the manufacturer's instructions (ThermoFisher Scientific). Samples were analyzed via flow cytometry on a BD LSR-FLOW Cytometer and analysis was performed with a minimum of 10,000 events.

## Flow cytometry

Mixed hiPSC populations and pure population controls were dissociated from tissue culture plates with Accutase (STEMCELL Technologies) and washed with PBS. Cells were fixed for 15 min with 4% paraformaldehyde (VWR) and washed 3X for 3 min with PBS. Samples were incubated in Hoescht stain (1:10,000) for 30 min and run on a LSR-II analyzer (BD Biosciences) to detect the ratio of WT-GFP(+) to CRISPRi mCherry(+) populations, as well as % of EdU +cells. Analysis was conducted with a minimum of 10,000 events per sample.

## FACS

Mixed hiPSC populations and pure population controls were dissociated from tissue culture plates and washed 3X with PBS. A LIVE/DEAD stain (ThermoFisher Scientific) was used per manufacture instructions. Prior to sorting, cells were suspended in PBS supplemented with Y-27632(10 µM) and kept on ice. A BD FACSAria II cell sorter (BD Biosciences) was used to isolate pure populations of WT-GFP and CRISPRi hiPSCs by first identifying the live cells via the LIVE/DEAD(350) stain and subsequently sorting the mCherry(+) GFP(-) populations from the mCherry(-)GFP(+) populations directly into TRIzol LS Reagent (ThermoFisher Scientifc). Samples were then stored at −80°C until subsequent mRNA extraction.

## Fluidigm 96.96 array

Sorted hiPSCs stored in TRIzol LS were thawed on ice and mRNA was extracted using a Direct-zol RNA MiniPrep Plus kit (ZYMO Research) following the manufacturer's instructions. RNA was converted to cDNA using the iScript cDNA synthesis kit (Bio-Rad). Forward and reverse primers for genes were designed using NCBI's Primer-BLAST (*Supplementary file 1* - Table 3). Primers were pooled to 500 nM to enable specific-target amplification and cDNA was amplified with PreAmp Master Mix (Fluidigm) and pooled primers for 15 cycles. Pre-amplified samples mixed with 2X Sso-Fast EvaGreen Supermix with low ROX(Bio-Rad) and 20X DNA Binding Dye Sample Loading Reagent (Fluidigm), and 10 µM primer sets were mixed with 2X Assay Loading Reagent (Fluidigm). 5 µl of diluted cDNA and primers and were loaded onto the IFC chip per manufacturer's instructions and loaded into the chip using the IFC Controller HX (Fluidigm). qPCR was run for 40 cycles in the IFC chip using the BioMark HD in the BioMark HD Data Collection Software. Resulting data was analyzed in the Real-Time PCR Analysis Software. All instruments and software involved with the IFC chip were manufactured by Fluidigm. Gene expression levels were calculated with respect to time-matched pure populations of WT hiPSCs, and hierarchically clustered and plotted using Genesis software (Institute for Genomics and Bioinformatics, Graz University of Technology).

## Human iPSC differentiation

For the dual SMAD and CHIR germ lineage differentiations, 100 cell mixed colonies were generated as previously described, cultured in mTeSR™-1 medium (STEMCELL Technologies), and allowed to form patterns for 5 days in pluripotency maintenance conditions. After 5 days, the media was supplemented with either SB 435142 (10µM; Stemgent) and LDN 193189 (0.2µM; Sigma-Aldrich) or CHIR 99021 (12µM; Selleckchem). CHIR was pulsed for 24 hr periods on the first and fourth day of the mesendoderm directed differentiation. Dual SMAD inhibition was kept constant for 6 days by supplementing SB 435142 and LDN 193189 into MTeSR media to direct germ lineage to an ectodermal fate. After 6 days of differentiation, colonies were washed 3X with PBS and fixed for staining with 4% paraformaldehyde (VWR) as previously described.

## Computational image analysis

For the radial position computational analysis, fluorescent images were split into single RGB channel images using the Python module scikit-image (*scikit-image contributors et al., 2014*) where the red channel denoted CRISPRi cells, the green channel denoted WT cells, and the blue channel denoted DAPI staining of the entire colony. A mask of each channel was created by thresholding, removing small objects, and removing small holes. The radius (r) of each colony was calculated using the DAPI mask and the ratio of inner red cell area vs. outer red cell area was calculated by taking the logical AND of the red channel mask and the DAPI mask above 1/3 r vs the logical AND of the red channel mask and the DAPI mask below ¼ r and normalizing to the total red mask area. To ensure accuracy,

the inner vs. outer red ratio was averaged with the inverse of the inner vs. outer green ratio (calculated in the same manner using the green channel mask).

For the differentiation analysis, fluorescent images were split into single RGB channels where the red channel denoted CRISPRi cells (mCherry+), the green channel denoted WT cells (GFP+), and the blue channel denoted either PAX6 or EOMES positive cells. The pixel area of cell types was determined by thresholding the WT, CRISPRi, or EOMES/PAX6 +images before 'removing small objects' and 'removing small holes' to create a mask of the area covered by each individual cell type. The EOMES or PAX6 +population was calculated by taking the logical AND of either the WT and EOMES/PAX6 +mask or the CRISPRi and the EOMES/PAX6 +masks and then normalizing to the total area of the WT or CRISPRi mask, respectively. The ratio of EOMES/PAX6 +cells in CRISPRi cell compared to WT was calculated by dividing the normalized EOMES/PAX6 +area of the CRISPRi mask by the normalized EOMES/PAX6 +area of the WT mask.

## Statistics

To ensure unbiased sampling of colonies in all cell mixing experiments, 10 colonies were randomly chosen on day zero before pattern formation and imaged daily thereafter. Each experiment was performed with at least three biological replicates. Unpaired T-tests were used to compare two groups. One-way analysis of variance (ANOVA) was used to compare three or more groups, followed by post-hoc pairwise comparisons by Tukey's tests. In gene expression analysis, three replicates were used for each condition, and all gene expression normalized to control mixed populations (off-target guide without knockdown) to control for any gene expression changes due to mixing or the process of FACS sorting. In all comparisons, significance was specified as $p \leq 0.05$.

## Acknowledgements

We thank the Gladstone Light Microscopy and Histology Core, the Gladstone Flow Cytometry Core (NIH P30 AI027763, NIH S10 RR028962, and the James B Pendleton Charitable Trust), and the Gladstone Stem Cell Core for their support and experimental expertise. ARGL received support from the National Heart Lung and Blood Institute (NIH 1F31HL140907).BRC and TCM were supported by the Gladstone Institutes. TCM received support from the California Institute of Regenerative Medicine (LA1_C14-08015) and the National Science Foundation (CBET 0939511). BRC received support from the National Institutes of Health (U01HL100406, P01HL089707, R01HL130533). We are grateful to Ariel Kauss for providing critical analysis and feedback on this manuscript.

## Additional information

### Competing interests

Mohammad A Mandegar: Currently an employee of Tenaya Therapeutics. Bruce R Conklin: On the Scientific Advisory Boards of: CODA Therapeutics, making therapies related to pain relief; Cellogy Inc., a cell dynamics measurement company; and Scientist.com, a scientific services company. Also a founder of Tenaya Therapeutics (https://www.tenayatherapeutics.com/), a company focused on finding treatments for heart failure, including the use of CRISPR interference to interrogate genetic cardiomyopathies. Holds equity in Tenaya, and Tenaya provides research support for heart failure-related research. Todd C McDevitt: A consultant for Tenaya Therapeutics. The other authors declare that no competing interests exist.

### Funding

| Funder | Grant reference number | Author |
|---|---|---|
| National Institutes of Health | Center Core Grant, NIH P30 AI027763 | Ashley RG Libby |
| National Institutes of Health | Instrumentation Grant, NIH S10 RR028962 | Ashley RG Libby |
| James B. Pendleton Charitable Trust | | Ashley RG Libby |

| | | |
|---|---|---|
| National Heart, Lung, and Blood Institute | F31HL140907 | Ashley RG Libby |
| National Institutes of Health | U01HL100406 | Bruce R Conklin |
| National Institutes of Health | P01HL089707 | Bruce R Conklin |
| National Institutes of Health | R01HL130533 | Bruce R Conklin |
| Emergent Behaviors of Integrated Cellular Systems | NSF-CBET 0939511 | Todd C McDevitt |
| California Institute for Regenerative Medicine | LAI-C1408015 | Todd C McDevitt |

The funders had no role in study design, data collection and interpretation, or the decision to submit the work for publication.

## Author contributions

Ashley RG Libby, Conceptualization, Data curation, Formal analysis, Validation, Investigation, Visualization, Methodology, Writing—original draft, Writing—review and editing; David A Joy, Software, Formal analysis, Visualization, Writing—review and editing; Po-Lin So, Conceptualization, Supervision, Project administration, Writing—review and editing; Mohammad A Mandegar, Federico N Mendoza-Camacho, Data curation, Validation, Writing—review and editing; Jonathon M Muncie, Data curation, Formal analysis, Writing—review and editing; Valerie M Weaver, Supervision, Methodology, Writing—review and editing; Bruce R Conklin, Conceptualization, Resources, Supervision, Visualization, Methodology, Writing—review and editing; Todd C McDevitt, Conceptualization, Resources, Data curation, Software, Supervision, Funding acquisition, Investigation, Visualization, Methodology, Writing—original draft, Project administration, Writing—review and editing

## Author ORCIDs

Ashley RG Libby http://orcid.org/0000-0002-8139-8844
Todd C McDevitt https://orcid.org/0000-0002-8905-4931

## Decision letter and Author response

Decision letter https://doi.org/10.7554/eLife.36045.021
Author response https://doi.org/10.7554/eLife.36045.022

# Additional files

## Supplementary files

• Supplementary file 1. Supplementary tables. (1) Table 1 displays guide RNA sequences used for CRISPRi knockdown. (2) Table 2 outlines primer sequences used for gene expression analysis by quantitative PCR. (3) Table 3 shows gene targets and primer sequences used for the Fluidigm 96.96 array gene expression analysis.
DOI: https://doi.org/10.7554/eLife.36045.018
• Transparent reporting form
DOI: https://doi.org/10.7554/eLife.36045.019

## Data availability

All data generated or analyzed in this study are included in the manuscript or supplementary files.

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
