## [Decision Letter]

Thank you for submitting your article "Spatiotemporal mosaic self-patterning of pluripotent stem cells using CRISPR interference" for consideration by *eLife*. Your article has been reviewed by three peer reviewers, including Gordana Vunjak-Novakovic as the Reviewing Editor and Reviewer #3, and the evaluation has been overseen by Didier Stainier as the Senior Editor. The following individual involved in review of your submission has agreed to reveal their identity: Krishanu Saha (Reviewer #2).

The reviewers have discussed the reviews with one another and the Reviewing Editor has drafted this decision to help you prepare a revised submission.

Summary:

Libby and colleagues report the methodology for inducing differentiation of human iPS cells into multiple populations capable of forming complex multicellular structures, by genetic control of symmetry-breaking events in cultured cells. Two molecular regulators were knocked down: ROCK1 (regulating cortical tension) and CDH1 (regulating cell-cell adhesion). Interestingly, the co-culture of gene edited cells with wild type cells resulted in spatial patterning of cell populations while their pluripotency has been maintained. The importance of this methodology will likely go way beyond the scope of this study, as a powerful and highly controllable tool for studies of development, maturation and disease using human cell and tissue models. The study is well planned and executed, with a few areas that need clarification and improvement. In addition, the impact of the findings would be further increased by providing at least one demonstration of how cell patterning could, be used in studies of signal propagation, morphogenic events in organoid systems, or modeling of diseases.

Essential revisions:

1) Mixing of the cells is an important part of the methodology and the way it is done may have major effects on biological outcomes. The paper would benefit from more rigorous and more extensive evaluation of mixing (segregation) using images from multiple biological replicates and multiple independent experiments. Also, the authors should explain how the individual images were selected (randomly or not; with or without exclusion criteria), and how exactly they assessed the differences between the groups and time points. In some instances, the knockout and control samples look more similar than one would expect, and it will be important to eliminate any ambiguity in data assessment and interpretation.

2) While the methodology reported in this paper is clearly applicable to a number of biological questions, the authors do not provide examples (case studies) for such applications. If possible, an example illustrating the utility of cell patterning should be included, and this would further increase the impact and value of this excellent work. We hope you have such an analysis in hand and could include it here.

3) Although the CDH1 KD population appears to maintain OCT4/SOX2 expression during the mixing studies, the β-catenin staining appears to show nuclear localization of β-catenin in the CDH1 KD colonies. If the authors would like to claim that the CDH1 KD maintains pluripotency within the pure colonies (as they do in the manuscript based on morphology of the colonies), they should show expression of Wnt targets, along with quantified expression of pluripotency-associated proteins in the pure colonies after CDH1 KD. How do BRA (primitive streak) and SOX9 (neural crest) gene or protein expression look in these CDH1 KD colonies?

4) The observation that CDH1 expression is not significantly different in the CDH1 KD fraction in the mixing study at the 6-day timepoint when the cells are mixed with the control cells for the duration is interesting. Can the authors explain why this is happening? Also, the authors should comment on the multi-layer organization of the ROCK1 KD upon mixing with the WT population in Figure 2F.

In the control images shown by the authors where WT-GFP cells are mixed with the DOX treated CRISPRi line without any gRNA added, there already appears to be some segregation likely owing to differential cell-cell adhesion between the two subpopulations (Figure 2B, first row). This makes the interpretation somewhat difficult.

5) It is not entirely clear how the cell density (clearly an important factor in cell fate and function) was established and controlled in the experiments. Consideration should also be given to the changes in cell density over culture time, which may be different for different cell types. In Figure 2—figure supplement 1, the control CRISPRi cells in the 25% condition seem to be more than 25% of the overall population on Day four. Do the control CRISPRi cells dominate over the WT iPSCs? The proportion of CRISPRi cells to WT cells in the control, ROCK1 KD, CDH1 KD cases seem to be different at the end of Day four.

6) Throughout the paper, the statements about the exact numbers of replicates and experiments, and the reproducibility of the reported data should be included.

---

## [Author Response]

Essential revisions:1) Mixing of the cells is an important part of the methodology and the way it is done may have major effects on biological outcomes. The paper would benefit from more rigorous and more extensive evaluation of mixing (segregation) using images from multiple biological replicates and multiple independent experiments. Also, the authors should explain how the individual images were selected (randomly or not; with or without exclusion criteria), and how exactly they assessed the differences between the groups and time points. In some instances, the knockout and control samples look more similar than one would expect, and it will be important to eliminate any ambiguity in data assessment and interpretation.

We acknowledge that the mixing of two different cell populations requires rigorous evaluation and we apologize for the lack of clarity originally describing the number of biological replicates and how patterns were quantitatively classified. We have now included the number of biological replicates in each of the figure legends, as well as added a section to the Materials and methods describing how images were selected. Additionally, we quantified our results by developing computational image analysis tools in Python (Figure 2D, H) that enabled blinded analysis of all the images that were collected.

2) While the methodology reported in this paper is clearly applicable to a number of biological questions, the authors do not provide examples (case studies) for such applications. If possible, an example illustrating the utility of cell patterning should be included, and this would further increase the impact and value of this excellent work. We hope you have such an analysis in hand and could include it here.

The utility of patterning of pluripotent stem cells (PSCs) in a “bottom up” manner as we describe establishes a unique means by which the co-emergence of multiple cell populations can be predictably controlled as PSCs differentiate. To provide examples of this utility, we now include a supplemental figure (Figure 4—figure supplement 2) with the results of two directed differentiation protocols targeting different germ layers (ectoderm and mesendoderm using dual SMAD inhibition or transient CHIR treatment, respectively). In both cases, we immunostained the cells after 6 days for the lineage markers PAX6 and EOMES to assess relative differentiation to ectoderm and mesoderm. We observed that the ROCK1 KD cells in mixed colonies stained for both PAX6 and EOMES with no significant differences from the control. However CDH1 KD populations predominantly expressed nuclear EOMES over control populations and had fewer cells expressing PAX6. A discussion of the potential applications of these results has been included in the text.

3) Although the CDH1 KD population appears to maintain OCT4/SOX2 expression during the mixing studies, the β-catenin staining appears to show nuclear localization of β-catenin in the CDH1 KD colonies. If the authors would like to claim that the CDH1 KD maintains pluripotency within the pure colonies (as they do in the manuscript based on morphology of the colonies), they should show expression of Wnt targets, along with quantified expression of pluripotency-associated proteins in the pure colonies after CDH1 KD. How do BRA (primitive streak) and SOX9 (neural crest) gene or protein expression look in these CDH1 KD colonies?

We fully acknowledge that expression of OCT4/SOX2 alone is not sufficient to ensure pluripotency is being retained, but the coincident expression of both markers is consistent with maintenance of a pluripotent phenotype. To further address this concern, we performed additional qPCR on both CDH1 KD and ROCK1 KD cultures to examine lineage markers associated with the primitive streak (BRA) and the neural crest (SOX9) to supplement the quantitative analysis of pluripotent-associated markers. The results of these analyses are now included in a new panel added to Figure 3—figure supplement 2.

4) The observation that CDH1 expression is not significantly different in the CDH1 KD fraction in the mixing study at the 6-day timepoint when the cells are mixed with the control cells for the duration is interesting. Can the authors explain why this is happening? Also, the authors should comment on the multi-layer organization of the ROCK1 KD upon mixing with the WT population in Figure 2F.In the control images shown by the authors where WT-GFP cells are mixed with the DOX treated CRISPRi line without any gRNA added, there already appears to be some segregation likely owing to differential cell-cell adhesion between the two subpopulations (Figure 2B, first row). This makes the interpretation somewhat difficult.

We apologize for any confusion regarding the presentation of the data in Figure 4; this was a complex study with a lot of comparative controls. We have attempted to address this by restructuring the figure to better depict the data and provide a clearer explanation of the results in the text and figure legend. Additionally, we included a separate qPCR study specific for mixed colonies interrogating genes associated with pluripotency and early germ lineages (Figure 4C, D).

As the multilayer organization of the ROCK1 KD cells occurs only during short term KD, when KD is induced after mixing with WT, we have addressed these results in the Discussion by describing how ROCK inhibition studies compare to our results and the potential implications of long term knockdown of ROCK1 on cellular biology.

We acknowledge that some differential adhesion between cell populations cultured separately beforehand may occur upon mixing. In order to better address how well the different cell populations segregate, we now include quantitative data of radial cell distribution (Figure 2D, H) and description of pattern classification as mentioned above.

5) It is not entirely clear how the cell density (clearly an important factor in cell fate and function) was established and controlled in the experiments. Consideration should also be given to the changes in cell density over culture time, which may be different for different cell types. In Figure 2—figure supplement 1, the control CRISPRi cells in the 25% condition seem to be more than 25% of the overall population on Day 4. Do the control CRISPRi cells dominate over the WT iPSCs? The proportion of CRISPRi cells to WT cells in the control, ROCK1 KD, CDH1 KD cases seem to be different at the end of Day 4.

We recognize that over time cell density within a colony may change depending on what gene(s) might be KD. In order to address this, we performed additional studies to assess the proliferation rates of the different cell lines used in the studies. We included these results in a new supplemental figure (Figure 4—figure supplement 1) and have included text describing our observations in the Results section.

6) Throughout the paper, the statements about the exact numbers of replicates and experiments, and the reproducibility of the reported data should be included.

We apologize for not better highlighting the numbers of biological replicates within our studies and not conveying the number of experiments performed more clearly throughout the manuscript. As previously mentioned, we have now included a section in the Materials and methods that describes how many biological replicates were examined and also randomized. These details are now highlighted in the text and figure legends as well to ensure transparency and accurate reporting of our scientific rigor.